


# Demonstrating the use of UNSEEN climate data for hydrological applications: case studies for extreme floods and droughts in England

Kay, Alison L[1], Dunstone, Nick [2], Kay, Gillian[2], Bell, Victoria A[1], Hannaford, Jamie[1]

[1]UK Centre for Ecology & Hydrology, Wallingford, UK, OX10 8BB

[2]Met Office, Exeter, UK, EX1 3PB

*Correspondence to:* A Kay (alkay@ceh.ac.uk)

**Abstract.** Meteorological and hydrological hazards present challenges to people and ecosystems worldwide, but the limited length of observational data means that the possible extreme range is not fully understood. Here, a large ensemble of climate model data is combined with a simple grid-based hydrological model, to assess unprecedented but plausible hydrological extremes in the current climate across England. Two case studies are selected—dry (Summer 2022) and wet (Autumn 2023)—with the hydrological model initialised from known conditions then run forward for several months using the large climate ensemble. The modelling chain provides a large set of plausible events including extremes outside the range from use of observed data, with the lowest flows around 28% lower on average for the Summer 2022 drought study and the highest flows around 42% higher on average for the Autumn 2023 flood study. The temporal evolution and spatial dependence of extremes is investigated, including the potential time-scale of recovery of flows to normal and the chance of persistent extremes. Being able to plan for such events could help improve the resilience of water supply systems to drought, and improve flood risk management and incident response.

**Keywords.** River flows, extreme events, floods, droughts, hydrological modelling

## 1    Introduction

Meteorological and hydrological hazards — storms, floods and droughts — present challenges to people, infrastructure and ecosystems globally (Beevers et al. 2022; Mokhtari et al. 2023). The relatively limited period of available observational data means that the possible range of extremes of such events is often not fully understood. For example, Thompson et al. (2017) show that in SE England there is a 7% chance of exceeding the current observed rainfall record in at least one month in any given winter, with a 34% chance of breaking a regional record somewhere in England and Wales. Similarly, Kent et al. (2022) investigate plausible summer rainfall extremes, showing an approximately 1% chance per year of exceeding current daily rainfall records in the UK in the current climate. Chan et al. (2023b) show an approximately 9% chance of a summer month with lower rainfall than the observed driest summer in SE England in the current climate. When such unprecedented events do inevitably occur, they can lead to very severe impacts (Bertola et al. 2023). This is not just because of their unprecedented magnitude but due to inherent unpreparedness, given that water supply systems, flood infrastructure, and related risk management strategies are typically adapted to historical ranges of variability (Kjeldsen and Prosdocimi 2016,).

One possible way to assess how events may unfold is the use of a so-called 'storyline' approach; a storyline is typically defined as a "physically self-consistent unfolding of past events, or of plausible future events or pathways" (Shepherd et al. 2018). This approach has been advocated in the context of future climate change, as a


way of circumventing the deep uncertainties associated with future climate projections by placing more emphasis
on driving factors and plausibility than probability (Shepherd et al. 2018). In this context it has links to so-called
H++ scenarios, which are plausible but high-end scenarios of climate change (Reynard et al. 2017). But storylines
can equally be applied in the context of past events by developing plausible counterfactuals, i.e. alternative ways
that those events could have unfolded even in the current climate (Sillmann et al. 2021). For example, Chan et al.
(2022) developed storylines for the UK drought of 2010-2012, by applying changes to the observed event based
on i) antecedent conditions (applying progressively drier conditions), ii) temporal sequencing (adding a dry winter
before or after the observed event), and iii) climate change. The results showed the importance of hydrological
initial conditions, and the vulnerability of catchments in Britain to a 'third dry winter'. Such studies can aid
preparedness by enabling planning for events similar to, but more extreme than, known events (with known
responses and impacts). However, the development of storylines in this way requires expert judgment on
plausibility, and on the factors important to the development of a particular event. Possible spatial factors may
also be neglected; for example Chan et al. (2022) apply the same storylines for catchments across Britain, treating
catchments essentially independently.
The use of large ensembles of climate data can reduce the need for expert judgment and enable spatially consistent
analyses and estimation of likelihoods. The extreme winter rainfall study of Thompson et al. (2017) was based on
a large ensemble of high-resolution initialised global climate simulations (termed 'UNSEEN', UNprecedented
Simulated Extremes using ENsembles), thus "directly sampling more extreme cases than the available
observations, allowing the identification of unprecedented rainfall events to assess their likelihood in the real
world". Statistical modelling could also be used to estimate the probability of unprecedented rainfall from
observations, and this is increasingly done in practice in UK water resources management using stochastic
simulation (e.g. Dawkins et al. 2022). However, the use of a dynamical model is judged to better preserve physical
plausibility and spatial dependence (Thompson et al. 2017). Data from either could be used to drive hydrological
models, to enable subsequent assessment of potentially unprecedented hydrological extremes, but the likely better
representation of spatial structures in dynamical models is important if large or multiple (not independent)
catchments are being considered. Chan et al. (2023a) use a large ensemble of seasonal global model hindcast data
to drive catchment-based hydrological models for 16 catchments (plus a groundwater model for 10 boreholes) in
the Anglian region of England, and used the summer 2022 drought as a case study to explore plausible storylines
of development into 2023. Brunner and Slater (2022) show that pooling reforecast ensemble members of European
river flows enables more robust estimates of very extreme flood events (those occurring less than twice in 100
years), with reduced uncertainty bounds compared to observation-based estimates.
Here, an expanded version of the UNSEEN ensemble of Thompson et al. (2017) is used in combination with a
simple grid-based hydrological model for Great Britain, to assess unprecedented but plausible hydrological
extremes in the current climate. For hydrological modelling, the antecedent conditions (e.g. water stored in the
soil and groundwater) are an important factor in subsequent river flows. Thus two case studies are selected, one
very dry (Summer 2022, associated with a major national-scale drought) and one very wet (Autumn 2023,
associated with persistent and large-scale flooding), with the hydrological model initialised from known
conditions at the start of each case study, then run forward for a number of months using the large ensemble of
UNSEEN climate data. The case studies are used to illustrate the potential of the approach to provide
unprecedented but plausible temporal and spatial hydrological extremes.




## 2    Data and methods

### 2.1    The UNSEEN climate datasets

The UNSEEN dataset used in this study comes from two hindcast ensembles of the Met Office Decadal Prediction System version 3 (DePreSys3; Dunstone et al. 2016), which is based on the Hadley Centre global coupled model (GCM) HadGEM3-GC3 (Williams et al. 2018). The model has an atmospheric grid resolution of ~60 km in the midlatitudes, and an ocean resolution of 0.25°. The first ensemble is initialised every 1st November and the second is initialised every 1st May, and each has forty ensemble members (realisations) but they are run for different lengths of time:

- 16-month periods starting in each November from 1959 up to 2021;
- 11-month periods starting in each May from 1960 up to 2022.

Differences between ensemble members are solely due to natural variability (i.e. there are no changes to GCM structure or parameterisation). Monthly rainfall rates are provided (mm/day) on a lat-long grid, for a region covering (most of) the UK (Figure 1a). The data for each realisation in an n-month period are considered dependent.

The GCM rainfall data are processed via the two steps below.

1. *Bias-corrected using simple monthly factors*. The 60km grids of correction factors are derived for each month of the Nov-initialised and May-initialised data separately (i.e. there are 16 resulting grids of correction factors for the Nov-initialised runs and 11 for the May-initialised runs), by comparing the GCM monthly mean precipitation across all initialisation-years and realisations against monthly mean observed precipitation from HadUK-Grid 1km data for 1961-2020 (Met Office et al. 2021) averaged up to the GCM grid. The grids of correction factors are then smoothed using a 3x3 grid around each cell, with a weight of 1/2 for the centre cell and 1/16 for each of the 8 surrounding cells (unless any surrounding cell has missing data, in which case its weight is added to that of the centre cell). A similar bias correction is derived for RCM precipitation data by Kay (2021). Maps of the correction factors are shown in Supp. Figures S1 and S2.

2. *Downscaled to the 1km GB national grid*. The hydrological model (Section 2.2) requires 1km inputs, which are derived from the ~60km lat-lon data by identifying the GCM grid box to use for each 1km grid box (Figure 1a) and distributing non-uniformly over the 1km grid boxes within each GCM grid box by using ratios of 1km to GCM grid box-mean standard average annual rainfall (SAAR) (Bell et al. 2007; Kay et al. 2023b). The SAAR ratios are shown in Figure 1b.

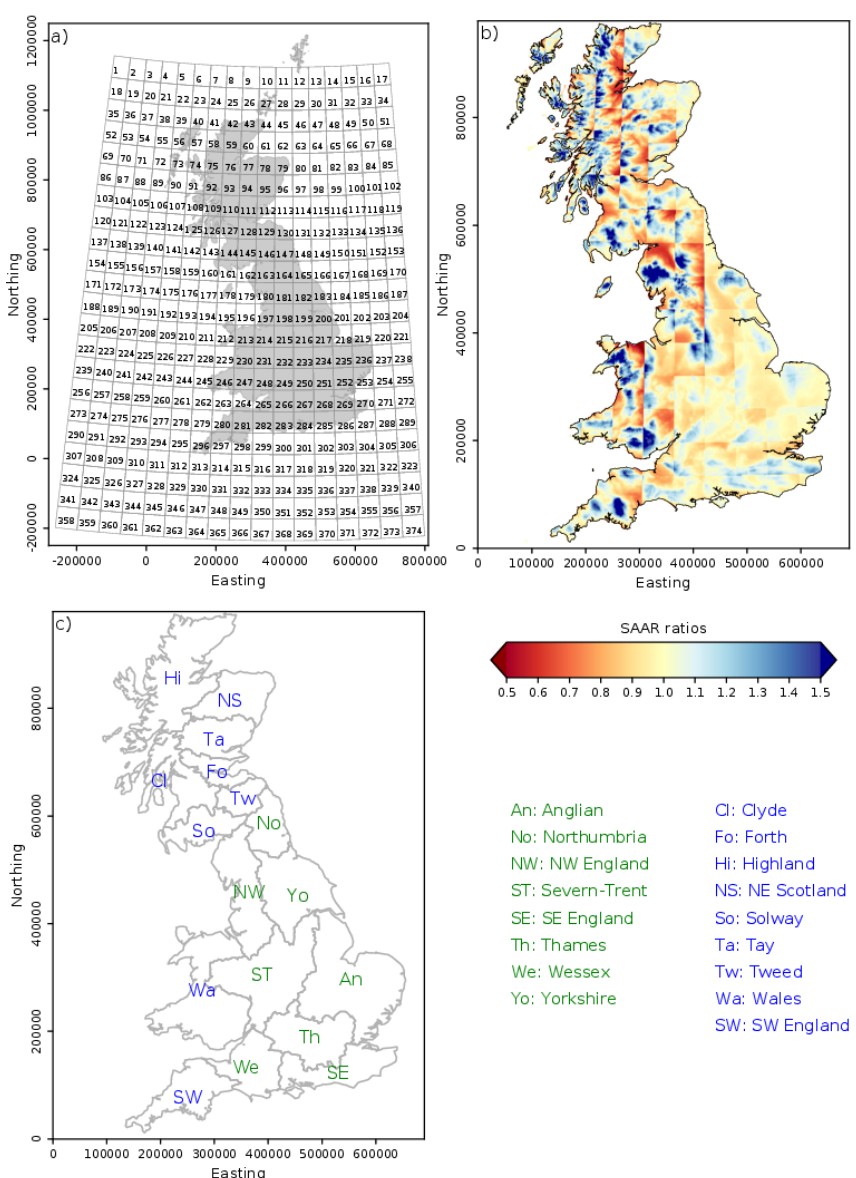

**Figure 1 a) Identifying GCM grid boxes on the GB national grid. b) The SAAR ratios used for downscaling GCM**
**rainfall to the 1km grid. c) The 17 UK Hydrological Outlook regions, with the eight regions used here in the left-hand**
**list (green) and the rest in the right-hand list (blue).**

## 2.2    The hydrological model

The Grid-to-Grid (G2G) is a grid-based runoff-production and routing model, operating on a 1km grid at a 15-
minute time-step across Great Britain (Bell et al. 2009). It is used for operational flood forecasting for England,
Wales and Scotland (Price et al. 2012; Cranston et al. 2012), and has been used to estimate the potential future





impacts of climate change on river flows across Britain (e.g. Kay et al. 2023a). However, the short time-step of
the model (required for stability of the routing scheme given the 1km grid scale) means that runs take a not
insignificant amount of time. For a seasonal forecasting application, where a relatively large number of runs were
required using coarse (temporal and spatial) climate data, a simple monthly Water Balance Model (WBM) was
developed, based upon G2G (Bell et al. 2013, 2017).
The WBM uses data from long historical runs of G2G (e.g. the long-term mean of the min and max subsurface
water storage depths, and the long-term mean monthly actual evaporation (AE)) as well as information on the
network of flow paths used by G2G, and is initialised using an estimate of subsurface water storage on a 1km grid
across GB, also taken from G2G for the required data. The WBM forms one component of the UK Hydrological
Outlook (UKHO; Prudhomme et al. 2017, hydoutuk.net, ukho.ceh.ac.uk), where it is initialised using a G2G
estimate of subsurface water storage derived using the most recent observations of rainfall and potential
evaporation (PE), and then run forward driven by an ensemble of Met Office rainfall forecasts for 1- and 3-months
ahead, to provide forecasts of regional mean river flows.
Bell et al. (2017) provide an assessment of the performance of the WBM compared to G2G, driving both with
observed 5km gridded rainfall data (1962-2010) and initialising the WBM from G2G at the start of each month.
Regional means of standardised 1km river flows for 17 regions across GB, for 1- and 3-months ahead, show
correlations of over 0.8 in all cases.
**2.3    Applying the climate data – two case studies**
As described in Section 2.2, the WBM needs to be initialised using an estimate of subsurface water storage on a
1km grid across GB, which is generally taken from a run of G2G driven by observed rainfall and PE data. Here,
the WBM is initialised for two case study events;
1.  Summer 2022 drought, and
2.  Autumn 2023 flood.
For each case study, the G2G model is run with daily observed data up to the end of the previous month, to produce
the initial conditions used by the monthly WBM. The observed data consists of 1km HadUK-Grid precipitation
(Met Office et al. 2021) and MORECS PE (Hough and Jones 1997). Then the WBM is run forwards for a number
of months using both UNSEEN ensembles for all realisations and all initialisation-years (1961-2022), regardless
of the year in which the UNSEEN climate data was initialised (hereafter 'WBM UNSEEN', Table 1).
For each case study, the WBM is also run forwards for the same period using all years of historical HadUK-Grid
1km precipitation data (1961-2022; hereafter 'WBM Obs'). This is analogous to the 'Ensemble Streamflow
Prediction' method used as part of the UKHO (Harrigan et al. 2018).
The use of UNSEEN (and observed) data from all years allows the widest examination of possible subsequent
rainfall pathways following summer 2022 and autumn 2023. The relatively long lead-times of the UNSEEN data
used here (Table 1), and hence the relatively weak predictable UK rainfall signals, results in a large dataset with
a wide range of possible outcomes including well-sampled dry (after summer 2022) and wet (after autumn 2023)
extreme tails. The WBM Obs ensemble provides a 'benchmark' that enables an assessment of the added-value of
the WBM UNSEEN ensemble.




**Table 1 The time-periods covered by the Nov- and May-initialised UNSEEN climate ensembles, and their use in each**
**case study. The grey boxes indicate UNSEEN data months regarded as spin-up. The green vertical lines indicate the**
**WBM initialisation for each case study (end-July 2022 for the summer 2022 drought study and end-October 2023 for**
**the Autumn 2023 flood study).**

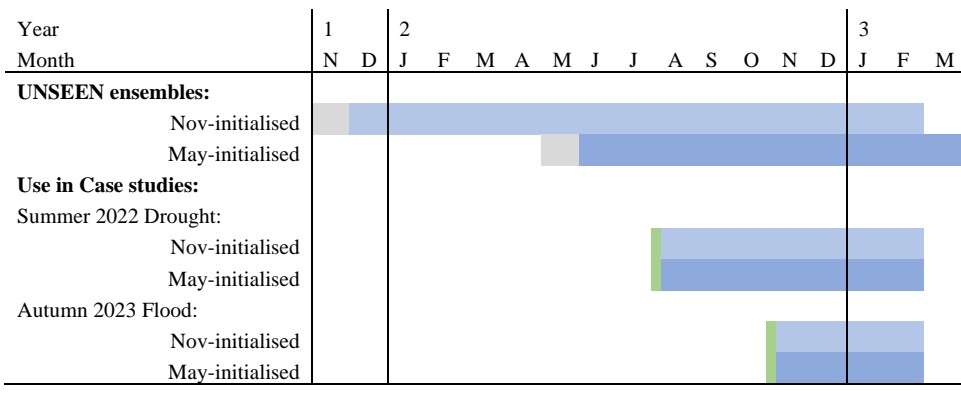

| Year | 1 | | 2 | | | | | | | | | | | | 3 | | |
|---|---|---|---|---|---|---|---|---|---|---|---|---|---|---|---|---|---|
| Month | N | D | J | F | M | A | M | J | J | A | S | O | N | D | J | F | M |
| **UNSEEN ensembles:** | | | | | | | | | | | | | | | | | |
| Nov-initialised | | | | | | | | | | | | | | | | | |
| May-initialised | | | | | | | | | | | | | | | | | |
| **Use in Case studies:** | | | | | | | | | | | | | | | | | |
| Summer 2022 Drought: | | | | | | | | | | | | | | | | | |
| Nov-initialised | | | | | | | | | | | | | | | | | |
| May-initialised | | | | | | | | | | | | | | | | | |
| Autumn 2023 Flood: | | | | | | | | | | | | | | | | | |
| Nov-initialised | | | | | | | | | | | | | | | | | |
| May-initialised | | | | | | | | | | | | | | | | | |

### 2.3.1 Summer 2022 drought

July 2022 was exceptionally warm and dry in the UK, with widespread record-breaking maximum temperatures
and extremely low rainfall across England (<10% of average over large parts of the south-east) (Barker et al.
2024). The preceding months had also been dry, leading to river flows and reservoirs reaching exceptionally low
levels across much of England and the introduction of temporary use bans by several water companies in southern
England (Barker et al. 2024).
We here ask the question 'how much worse might the situation have become'? To do this, the WBM is initialised
from G2G sub-surface conditions from end-July 2022 and run to end-Feb 2023 (7 months), with every UNSEEN
initialisation-year and realisation, and with every observed data year. This gives 4960 WBM UNSEEN runs for
each month (2 ensembles * 62 initialisation-years * 40 realisations), vs only 62 historical sequences for WBM
Obs, hence enabling a more robust assessment of rare low extremes.

### 2.3.2 Autumn 2023 flood

October 2023 was exceptionally wet across most of England, with rainfall totals exceeding 250% of average in
some areas, and the preceding three months had also been relatively wet (Hannaford et al. 2023). This led to river
flows reaching notably high levels across much of England, with elevated flood risk in many areas.
We again ask the question 'how much worse might the situation become'? In a similar way to the Summer 2022
analysis, the WBM is initialised from G2G sub-surface conditions from end-Oct 2023 and run to end-Feb 2024
(4 months), with every UNSEEN initialisation-year and realisation, and with every observed data year. This again
gives 4960 WBM UNSEEN runs for each month, vs only 62 for WBM Obs, hence enabling a more robust
assessment of rare high extremes.




### 2.4 Flow analyses

The WBM provides monthly mean river flows on a 1km grid, but in the UKHO these are typically standardised (by dividing by long-term mean flow from a set of observation-driven runs) and the standardised flows are displayed on a 1km grid or averaged across 17 regions (Figure 1c). This WBM approach was originally developed to provide an indication of the relative magnitude of monthly and regional mean river flows across GB using spatially-coarse GloSEA5 rainfall forecasts (Bell et al. 2017). A very similar approach is used here; although the DePreSys3 hindcasts have an improved spatial resolution, they are still relatively coarse, and focusing on regional mean flows greatly simplifies subsequent analyses and plotting. The results focus on eight regions of England; NW England, Northumbria, Severn-Trent, Yorkshire, Thames, Anglian, Wessex, SE England (Figure 1c).

### 2.4.1 Fidelity tests

Fidelity tests on WBM flow estimates are performed in a similar way to that applied for the UNSEEN rainfall data (Thompson et al. 2017 Fig. 2). That is, for each case study and region,

- the WBM UNSEEN (May- and Nov-initialised) simulated flows are resampled 1000 times, with each sample randomly selected from the 40*2 available realisations for the year, producing a time-series of the same length as the WBM Obs time-series;
- for each resample, the mean, standard deviation, skewness and kurtosis statistics are calculated;
- the same statistics are calculated for the WBM Obs simulated flows;
- if, for all four statistics, the WBM Obs flow value sits within the 2.5%-97.5% range of the values from the 1000 resamples of the WBM UNSEEN simulations, then the WBM UNSEEN simulations are considered to have passed the test (i.e. the distributions of WBM Obs and WBM UNSEEN flow values for the region and month are considered indistinguishable). If the test is only failed for one of the four statistics, then this is noted.

Passing the fidelity tests gives confidence in the ability of the WBM and UNSEEN GCM precipitation to simulate appropriate distributions of monthly mean river flows in a region, given the WBM initialisation for each case study.

### 2.4.2 Extreme flows

For each case study, the monthly time-series of regional mean flows from the full set of years and realisations of WBM UNSEEN are plotted as the median and range (5th-95th percentiles, and overall min and max). The Nov- and May-initialised ensembles are combined together. For comparison, the median and range from WBM Obs are also shown.

For historical context, the time-series of monthly regional mean flows for the previous two years are shown for each region, along with long-term mean flow ranges derived for the preceding years (from 1963). These are derived from runs of the WBM driven by observed data for 1963-2023, with initialisation (using G2G data) at the start of every month (hereafter 'WBM Obs 1m'). For each region, long-term mean flow ranges are derived by extracting the 5th, 13th, 28th, 72th, 87th and 95th percentiles for each month, along with overall min and max. These percentiles are chosen to match those defining the seven classes used in the UKHO: 'Exceptional low' (<5th),



'Notably low' (5th-13th), 'Below normal' (13th-28th), 'Normal' (28th-72th), 'Above normal' (72th-87th), 'Notably
high' (87th-95th), 'Exceptionally high' (>95th) ([ukho.ceh.ac.uk](ukho.ceh.ac.uk)).
The 'WBM Obs 1m' run described above is not directly comparable to the 'WBM UNSEEN' and 'WBM Obs'
ensemble runs, since the former is re-initialised from G2G for every month whereas the latter are only initialised
at the start of each case study then run forward for the $n$-months of each study (where $n$ is 7 for the Summer 2022
drought and 4 for the Autumn 2023 flood). To demonstrate any difference this may make, a further run is done
using observed driving data for each case study but only initialising at the start then running forward for $n$ months
(hereafter 'WBM Obs n-m').

### 2.4.3    Temporal and spatial variation in extreme flows

The UNSEEN-derived extreme flows are assessed, for each case study, by selecting and plotting the ensemble
member giving the most extreme flows for each month of the simulation, and investigating how these vary
temporally (for consecutive months) and spatially (for neighbouring regions). The UNSEEN ensemble member
is defined by the ensemble (Nov or May), the initialisation-year, and the realisation number (out of 40); see
Section 2.1.

### 2.4.4    Recovery or persistence of extreme flows

The long term mean flow bands (Section 2.4.2) are used to assess the chance of flows recovering to 'normal' by
each month, in each region, for each case study. For example, for the Summer 2022 drought case study, in each
region each ensemble member is assessed to see whether the flows have reached the 'normal' flow band (or
higher) by month $m$ but are still in a lower flow band for each month of the simulation prior to month $m$. Similarly,
for the Autumn 2023 flood case study, in each region each ensemble member is assessed to see whether the flows
have reached the 'normal' flow band (or lower) by month $m$ but are still in a higher flow band for each month of
the simulation prior to month $m$. The number of ensemble members recovering to 'normal' is then expressed as a
percentage of the full ensemble, and plotted for each region and each case study. In a similar way, the chance of
flows remaining at least exceptionally low (high) is assessed for Summer 2022 (Autumn 2023).

### 3    Results

### 3.1    Fidelity tests

The results of the flow fidelity tests are summarised in Table 2 for the each case study, indicating a pass ('1') or
fail (<=0) for each month and region, with a negative number indicating which statistic the test failed on if it only
failed on one of the four. These show overall pass-rates of 86% and 78% for the Summer 2022 drought and
Autumn 2023 flood events respectively. Note that there are no failures only on the mean ('-1'), because of the
correction applied to GCM precipitation data for mean monthly rainfall (Section 2.1).
Most of the failures occur in February; the pass-rates leaving out February are 96% and 96% respectively. The
failures in February are only related to the standard deviation ('-2'), which is due to this being too low for rainfall
in the climate model runs (Kelder et al. 2022). This is partly due to the extremely wet Feb 2020 (Davies et al.
2021); removing this from the February fidelity testing leads to passes in 6 out of 8 regions for both the Summer




1 2022 drought and Autumn 2023 flood (compared to only 2 when Feb 2020 is included). In each case, the test still

2 fails in Thames and SE England.

**Table 2 The results of fidelity tests for the Summer 2022 drought (left) and Autumn 2023 flood (right) case studies. '1'**

**indicates a PASS for all four statistics, while <=0 indicates a FAIL for at least one statistic. If there is only a FAIL on**

**one statistic then the negative number indicates which (-1 mean, -2 sd, -3 skewness, -4 kurtosis).**

| | Summer 2022 drought | | | | | | | Autumn 2023 flood | | | |
|---|---|---|---|---|---|---|---|---|---|---|---|
| | Aug | Sep | Oct | Nov | Dec | Jan | Feb | Nov | Dec | Jan | Feb |
| NWEngland | 1 | 1 | 1 | 1 | 1 | 1 | -2 | 1 | 1 | 1 | -2 |
| Northumbria | 1 | 1 | 1 | 1 | 1 | 1 | -2 | 1 | -3 | 1 | -2 |
| Severn-Trent | 1 | 1 | 1 | 1 | 1 | 1 | -2 | 1 | 1 | 1 | -2 |
| Yorkshire | 1 | 1 | 1 | 1 | 1 | 1 | -2 | 1 | 1 | 1 | -2 |
| Thames | 1 | 1 | 1 | 1 | 1 | 1 | -2 | 1 | 1 | 1 | -2 |
| Anglian | 1 | 1 | -4 | 1 | 1 | 1 | 1 | 1 | 1 | 1 | 1 |
| Wessex | 1 | 1 | 1 | 1 | 1 | 1 | 1 | 1 | 1 | 1 | 1 |
| SEEngland | 0 | 1 | 1 | 1 | 1 | 1 | -2 | 1 | 1 | 1 | -2 |

### 3.2 Extreme flows

Plots of the median and range of regional mean flows are shown in Figure 2 for the Summer 2022 drought case
study, and Figure 3 for the Autumn 2023 flood case study. These show that using the large ensemble of UNSEEN
data gives more extreme flows than using all of the historical observed data (red dotted lines vs grey dotted lines),
although the $5^{th}$, $50^{th}$ and $95^{th}$ percentiles from WBM UNSEEN and WBM Obs are all similar (orange dashed and
dot-dashed vs grey dashed and dot-dashed), as expected from the fidelity tests. It should be emphasised that, for
both WBM UNSEEN and WBM Obs, the min and max (dotted lines) represent the overall envelope of the
ensemble of simulations for each month, so they do not necessarily represent a plausible monthly evolution of the
flows (see Section 3.3). On average across the 8 regions, for the 7 months of the Summer 2022 drought case study
the low envelope of the WBM UNSEEN flows is 28% lower than that of the WBM Obs flows, and for the 4
months of the Autumn 2023 flood case study the high envelope of the WBM UNSEEN flows is 42% higher than
that of the WBM Obs flows.
Figure 2 also shows that the simulations from WBM Obs 1m (black solid line; initialised at the start of every
month) and WBM Obs n-m (black dashed line; only initialised at the start of the case study period) are very
similar. However, Figure 3 shows that the WBM Obs n-m simulation tends to over-estimate high flows relative
to the WBM Obs 1m re-initialised simulation; see Section 4.2 of Discussion.
The effect of initialisation is clear in all regions, but longer-lasting in some. For example, for the Summer 2022
drought study the conditions in July 2022 are obviously very dry (below normal in all regions, and exceptionally
low in some). In Severn-Trent the max from WBM Obs (top grey dotted) has increased from below the long-term
max to match it by Feb 2023, whereas in Thames the WBM Obs max is still well below the long-term max by
then. In all regions though, the min from WBM Obs (lower grey dotted) stays below the long-term min.
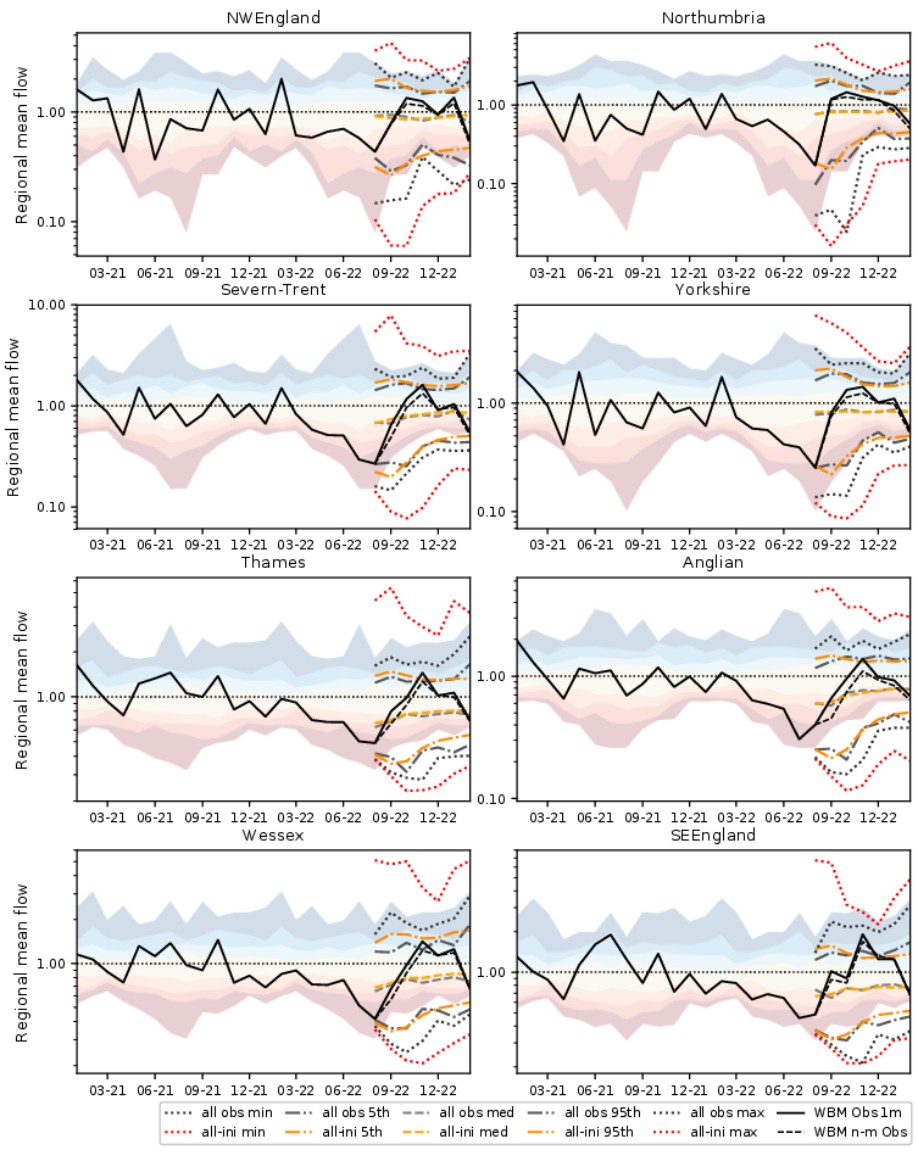

**Figure 2 Regional mean flows for the Summer 2022 drought case study. The dashed/dot-dashed/dotted lines show the median/5th-95th/min-max across the ensemble using all the observed driving data (grey) and all the UNSEEN driving data (orange/red) for the WBM initialised from end-July 2022. The WBM flows driven by observed data for Jan 2021-Feb 2023 and initialised at the start of every month are also shown (WBM Obs 1m; solid black line), as are the WBM flows driven by observed data for Aug 2022-Feb 2023 but only initialised at the start (WBM Obs n-m; dashed black line). For historical context, the coloured areas show the ranges from WBM Obs 1m for 1963-2020: min, 5th, 13th, 28th, 72nd, 87th, 95th, max percentiles. Note the log-scale on the y-axis, to emphasis lower flows.**



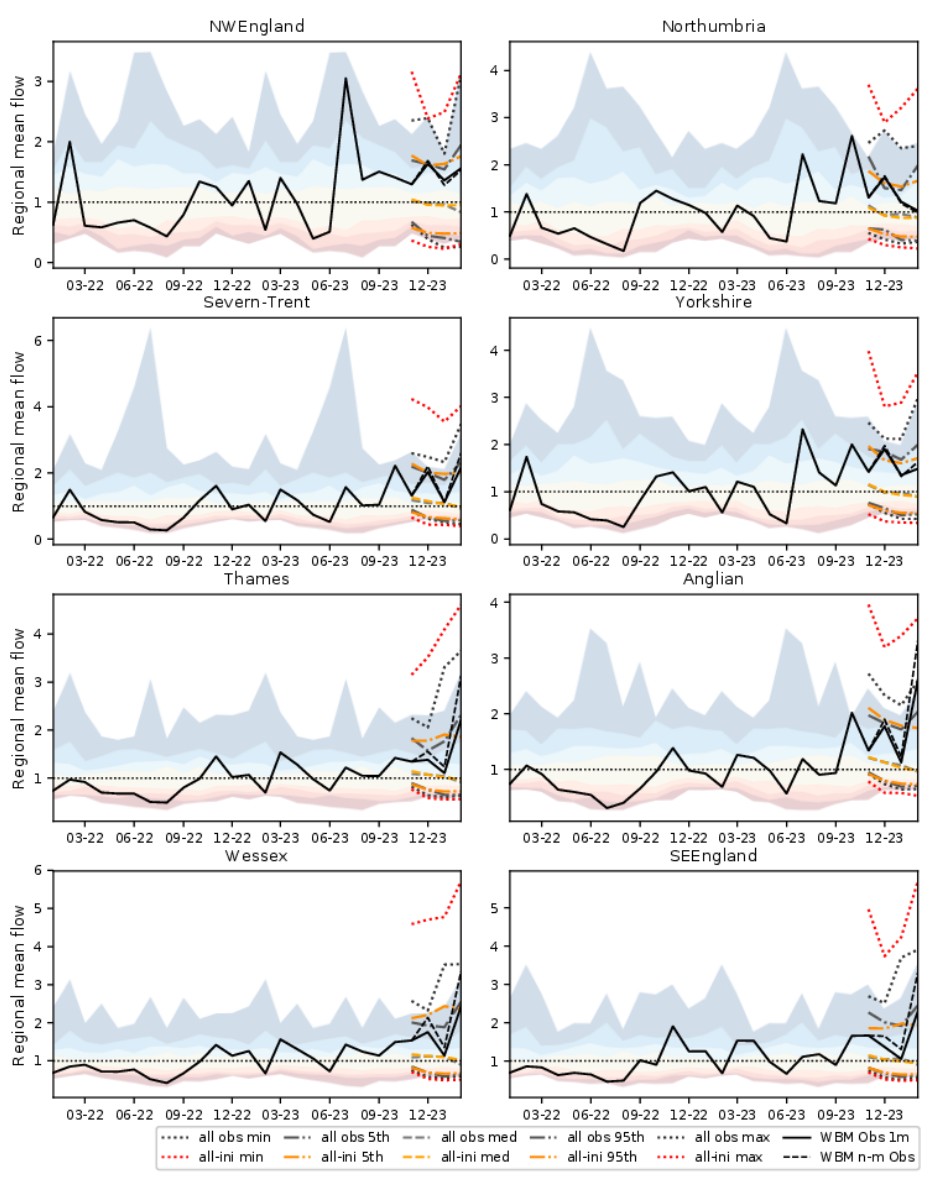

**Figure 3 As Figure 2 but for the Autumn 2023 flood case study. For historical context, the coloured areas show the**

**ranges from WBM Obs 1m for 1963-2021: min, 5th, 13th, 28th, 72th, 87th, 95th, max percentiles.**

Similarly, for the Autumn 2023 flood study the conditions in October 2023 are very wet (above normal in all

regions, and exceptionally high in some). In Northumbria the max from WBM Obs (top grey dotted) has decreased

from above the long-term max to match it by Dec 2023, whereas in Anglian the WBM Obs max is still well above

the long-term max even by Feb 2024. This is due to the presence of significant groundwater stores in regions to

the south/east of England, which typically respond much more slowly to weather conditions than the shallower





stores typically found to the north/west of England (Svensson et al. 2015). The WBM has information about the
spatial differences in sub-surface stores from the data it takes from long historical runs of G2G (Section 2.2).
**3.3      Temporal and spatial variation in extreme flows**
It is important to note that the WBM UNSEEN extremes for consecutives months are often not given by the same
UNSEEN climate ensemble member (ensemble, initialisation-year and realisation; Section 2.1). Similarly, the
extremes for different regions, even neighbouring ones, are often not given by the same ensemble member. This
is illustrated for four regions for the Summer 2022 drought (Figure 4) and for the Autumn 2023 flood (Figure 5).
Similarly, the WBM Obs extremes for consecutives months will likely not be given by the same ensemble
member.
For the Summer 2022 drought, Figure 4 shows that the ensemble members giving the lowest flows in August,
September or October 2022 each have a fast recovery in flows through the autumn. In contrast, the ensemble
members giving the lowest flows in late autumn and early winter 2022 are often low (but not lowest) earlier in
the year too, particularly in the Thames and Anglian regions, which are more influenced by slower-responding
groundwater systems. In no cases is the ensemble member giving the lowest flows for one month also that giving
the lowest flows for the following month. In only three cases does the ensemble member giving the lowest flows
in a given month and region also give the lowest flows for that month in a neighbouring region (Sep and Oct 2022
in Thames and Anglian, and Jan 2023 in Thames and Severn-Trent).
For the Autumn 2023 flood, Figure 5 shows that in no cases is the ensemble member giving the highest flows for
one month also that giving the highest flows for the following month, and only rarely is the ensemble member
giving the highest flows in a given month also very extreme in earlier/later months. The only real exception is the
ensemble member which gives the highest flows in Anglian in November 2023, which also gives flows higher
than previous observed records (although not the highest from the UNSEEN ensemble) for Dec 2023 and Jan
2024, falling back into the 'exceptionally high' range in Feb 2024. In only one case does the ensemble member
giving the highest flows in a given month and region also give the highest flows for that month in a neighbouring
region (Dec 2023 in Anglian and Severn-Trent).
These results also need to be interpreted in the context of the fidelity test results. In particular, in most regions the
fidelity tests were failed in February (Table 2), although this was mostly related to the extreme February observed
in 2020 so it may be that the results can otherwise be seen as representative. The fidelity tests were also failed in
the first month (August) of the Summer 2022 drought study in SE England, so those results may be less reliable.
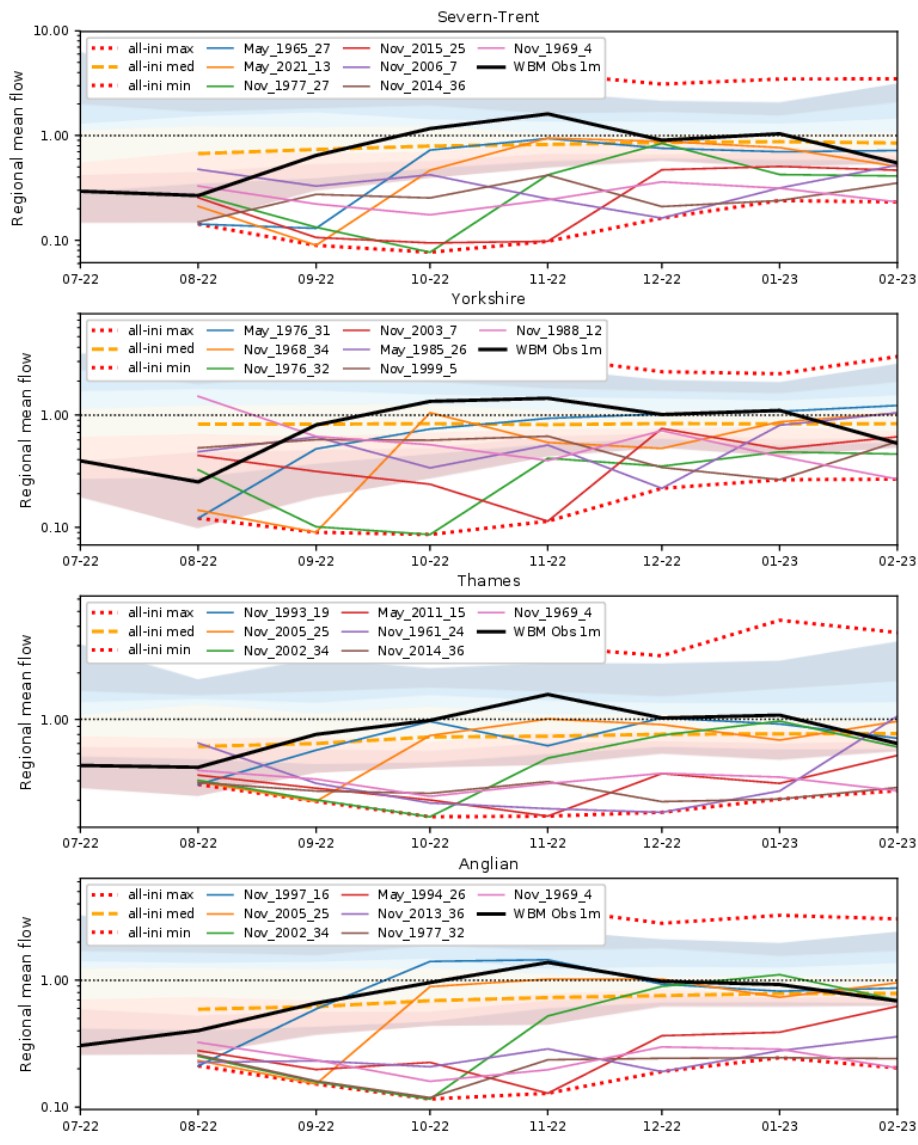

**Figure 4 Regional mean flows for the Summer 2022 drought case study, for the ensemble member giving the lowest flow in each month (Aug 2022 to Feb 2023; coloured solid lines, with labels identifying the ensemble member by ensemble (Nov or May), initialisation-year and realisation number). As in Figure 2, the dashed/dotted lines show the median/min-max across the ensemble using all the UNSEEN driving data (orange) for the WBM initialised from end-July 2022, and the WBM flows driven by observed data for Jul 2022-Feb 2023 and initialised at the start of every month are also shown (WBM Obs 1m; solid black line). For historical context, the coloured areas show the ranges from WBM Obs 1m for 1963-2020: min, 5th, 13th, 28th, 72nd, 87th, 95th, max percentiles.**

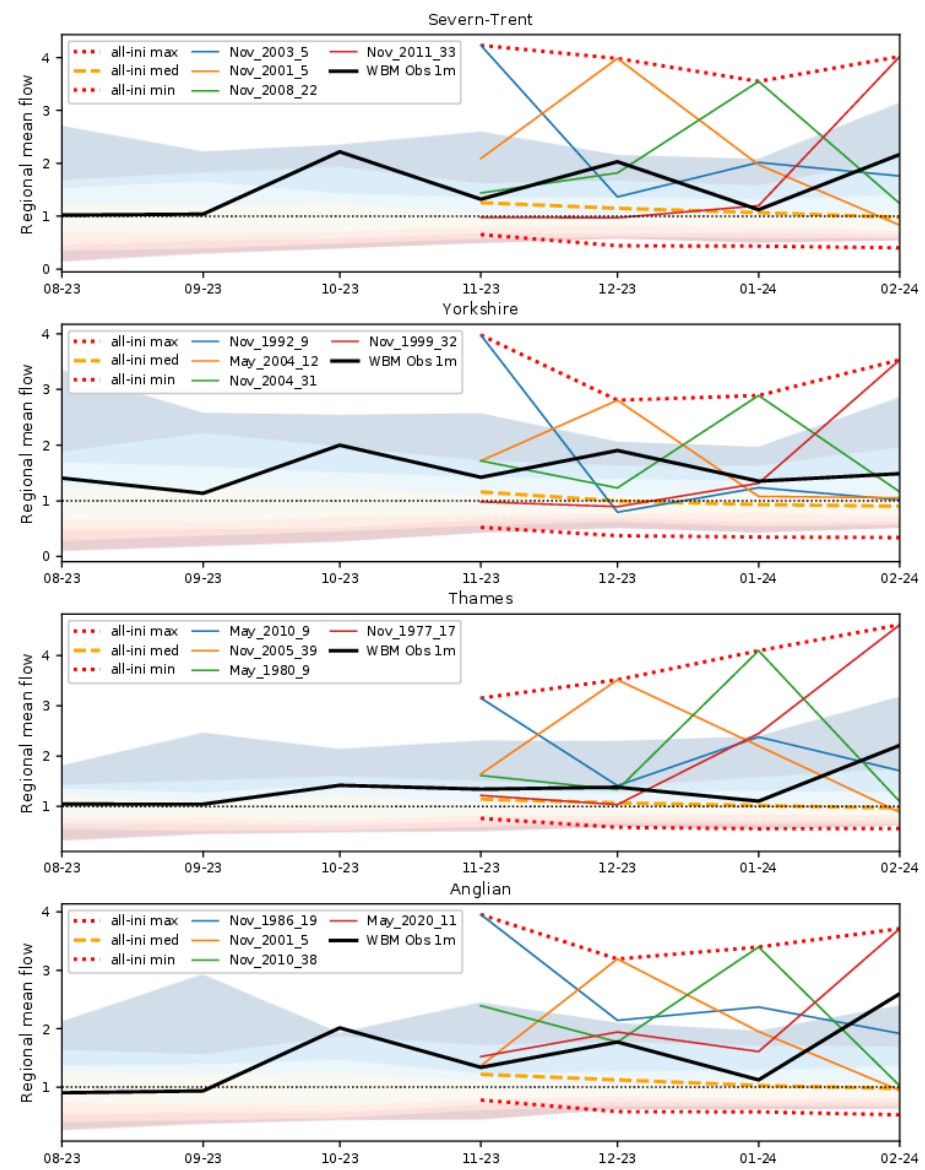

**Figure 5 As Figure 4 but for the Autumn 2023 flood case study. For historical context, the coloured areas show the ranges from WBM Obs 1m for 1963-2021: min, 5th, 13th, 28th, 72th, 87th, 95th, max percentiles.**

### 3.4 Recovery or persistence of extreme flows

The chance of flows recovering to 'normal', or remaining exceptionally low (or lower), by each month in each region for the Summer 2022 drought is shown in Figure 6. Flows in more northerly regions (NW England, Northumbria, Yorkshire) were very likely to have recovered to 'normal' by early 2023, whereas other regions show a slower recovery (from a lower starting point) — Anglian shows only around an 80% chance of recovery by early 2023 (Figure 6a). Conversely, the analysis of persistence of extremes shows around a 3% chance of flows




remaining exceptionally low in early 2023 in the Anglian region, with a zero chance of persistent low flows in
northerly regions (Figure 6b).

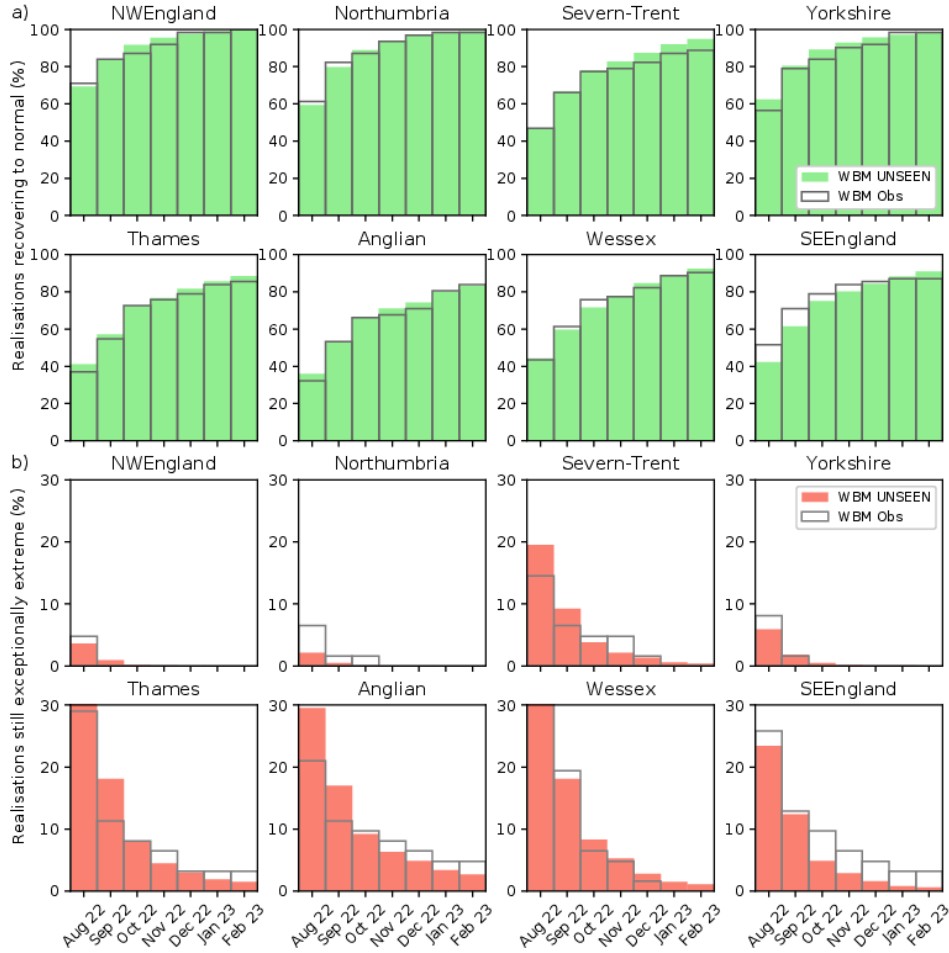

**Figure 6 The percentage of ensemble members showing a) recovery of flows to (at least) normal by each month, or b)**
**remaining at least exceptionally extreme by each month, for the Summer 2022 drought case study.**
For the Autumn 2023 flood, flows in regions to the north are very likely to have recovered to 'normal' by early
2024, whereas flows in regions to the south/east show only around a 70-85% chance of recovery by then, with
Anglian and Severn Trent worst (Figure 7a). Conversely, the analysis of persistence of extremes shows around a
3% chance of flows remaining exceptionally high (or higher) in early 2023 in the Wessex region, with a zero
chance of persistent high flows in the northerly regions (Figure 7b).
For both case studies, there are differences between the monthly percentages of recovery and persistence estimated
from the WBM UNSEEN ensemble and the WBM Obs ensemble (coloured bars vs outlined bars). The percentages
derived from the WBM UNSEEN ensemble appear to vary more smoothly from month to month than those
derived from the much smaller WBM Obs ensemble, and the larger ensemble size means that the rarer persistent


extremes should be estimated more robustly. The general patterns for recovery to normal flows are relatively
similar, but the patterns for persistence of exceptionally extreme flows are more different. In some regions there
are persistent extremes in the WBM UNSEEN ensemble that do not exist at all in the WBM Obs ensemble (e.g.
Wessex for the Summer 2022 drought lasting into 2023, and Thames and Severn-Trent for the Autumn 2023 flood
lasting into 2024), but in other regions the opposite is the case (e.g. SE England for the Summer 2022 drought
lasting into 2023, and Yorkshire for the Autumn 2023 flood lasting into 2024).

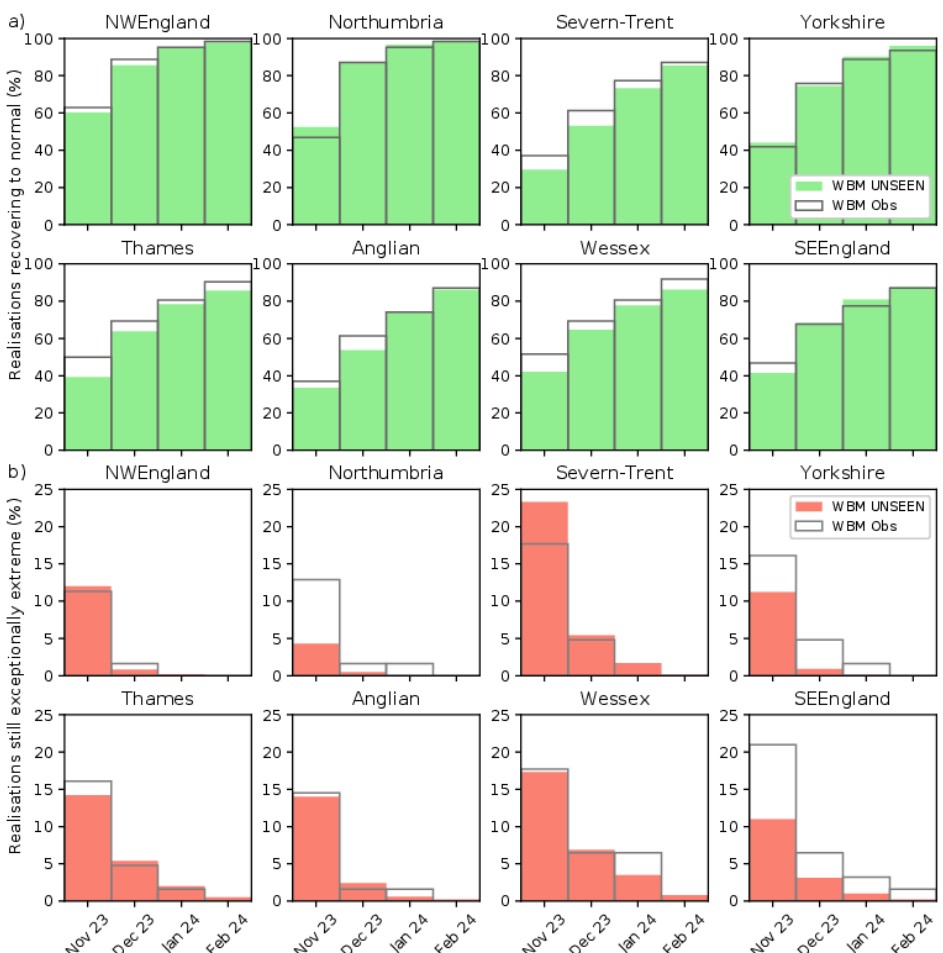

**Figure 7 As Figure 6 but for the Autumn 2023 flood case study.**
**4    Discussion**
**4.1    Case study outcomes**
Both the flood and drought case studies demonstrate the potential of using the large ensemble of UNSEEN climate
data in combination with a simple grid-based national-scale hydrological model. The modelling chain provides a
large set of plausible events including extremes outside the range from use of observed data, with the lowest flows



around 28% lower on average for the Summer 2022 drought study and the highest flows around 42% higher on
average for the Autumn 2023 flood study. It enables the investigation of the temporal evolution and spatial
dependence of extremes, including the potential time-scale and speed of recovery of flows to a normal range and
possible persistence of extremes across a number of months.
For the Summer 2022 drought case study, Figure 4 showed the existence of ensemble members giving very low
flows through to February 2023, particularly in the Thames and Anglian regions which are more influenced by
slower-responding groundwater systems. In reality, recovery of regional mean flows to normal happened
relatively quickly in autumn 2022 (black line), although there were localised drought concerns in parts of eastern
England into summer 2023, and concerns about a further drought developing more widely in early 2023 following
a dry winter (Sefton et al. 2023); some ensemble members show this possibility (e.g. realisation 19 from Nov
1993 in Thames – blue line in Figure 4).
Chan et al. (2023a) also used the summer 2022 drought as a case study to explore storylines of development of
extremes into 2023, but using a different seasonal hindcast dataset and only looking at selected catchments and
boreholes in the Anglian region of England. They split their large ensemble into four clusters (based on
atmospheric circulation indices), and showed that the clusters characterising drier-than-average winters resulted
in continuation of the drought into 2023. This highlights a further advantage of the use of large ensembles of
climate model data – the ability to characterise the large-scale drivers of extreme events.
For the Autumn 2023 flood case study, Figure 5 showed the ensemble member giving the highest flows in the
Anglian region in Nov 2023 also gave flows higher than previous records for Dec 2023 and Jan 2024, before
falling back into the 'exceptionally high' range in Feb 2024. Also, the ensemble member giving the highest flows
in Dec 2023 in Anglian gave the highest flows in the neighbouring Severn-Trent region. In reality, after a wet
December 2023 (Turner et al. 2024), severe flooding occurred over much of England very early in January 2024,
with over 250 flood warnings issued by the Environment Agency and over 1000 properties flooded
(floodlist.com/europe/united-kingdom/storm-henk-floods-january-2024). Flows then dropped back to some
extent later in January 2024, following drier weather across much of England (Sefton et al. 2024), before rising
again in February 2024, which was exceptionally wet across much of England leading to record high flows in
some catchments (nrfa.ceh.ac.uk/sites/default/files/HS_202402.pdf); some ensemble members show this
possibility (e.g. realisation 11 from May 2020 in Anglian – dark red line in Figure 5).
For each case study, the selection of ensemble members illustrating temporal and spatial variation (Section 3.3)
focussed on those that gave the most extreme flows for any given region and month. However, Section 3.4
included a summary of the percentage of ensemble members where flows remained at least exceptionally extreme
by each month. This illustrated a clear possibility of persistent extremes in some regions to the south/east, with
the chance of at least exceptionally low flows persisting from Summer 2022 into 2023 being ~3% in Anglian, and
the chance of at least exceptionally high flows persisting from Autumn 2023 into 2024 being ~3% for Wessex.
The selection of ensemble members could also focus on any giving extreme low/high flows across multiple
regions, to enable study of possible spatially extensive extreme events.



### 4.2 Limitations of the models and data

The WBM applied here is a very simple monthly hydrological model, which has both advantages and disadvantages. The monthly time-step of the model means that it runs very quickly, so can easily be used for large climate ensembles such as those applied here. However, it also assumes climatological AE (derived from long historical runs of G2G). The effect of this will likely be less in so-called water-limited areas to the south/east of England (where AE is generally limited by soil water availability, especially in summer) and in so-called energy-limited areas to the north/west of England (where AE is generally limited by PE), but it could have a larger effect in the more energy-water balanced areas in between (Kay et al. 2013), and also during more extreme wet/dry periods when AE should probably be higher/lower. This could be the reason for the possible over-estimation of high flows from the WBM Obs n-m run, relative to the WBM Obs 1m run which is re-initialised from G2G at the start of each month (Figure 3). Future work will investigate this possibility, and assess whether simple adjustments can be made to the WBM to improve simulation of extreme events.

The WBM also does not account for snow at all; although G2G has an optional snow module (Bell et al. 2016) it is not applied for the long historical runs used to provide data for the setup of the WBM, nor for the runs used for WBM initialisation. The lack of snow accounting will not have a significant effect for most regions of England most of the time, particularly given the monthly time-step of the WBM, as there are very few large and/or long-lasting accumulations of snow in most of England, although it is more important in parts of Scotland (Kay 2016).

The monthly time-step of the WBM, and the resolution of the UNSEEN data, is likely sufficient for investigating extreme low flows/droughts, which typically evolve relatively slowly due to rainfall deficits over extended periods of time. For extreme high flows/floods, a finer time-step would really be required as they can develop and recede much more quickly, particularly for smaller or flashier catchments. Despite this, the WBM can give a good indication of flood potential for most of England, because the main driver of floods here is soil moisture excess rather than extreme precipitation or snowmelt (Berghuijs et al. 2019). One potential approach could be to use a simple and fast-running model, like the WBM, to run a large ensemble, then select individual members of interest based on the outcomes from those runs. A much smaller set of runs of a more detailed model, like G2G, could then be performed using the particular members of interest to gain the extra temporal (and spatial) detail, provided (at least) daily precipitation data were available. Only monthly rainfall totals are available for the UNSEEN ensembles applied here, but future options will be investigated.

A simple bias correction was applied to the UNSEEN climate data before use to drive the hydrological model (Section 2.1). Applying this correction improved the results of the fidelity testing (not shown), although February in particular still has issues (related to the standard deviation of February rainfall being too low; Section 3.1). Bias-correction was also applied to climate data prior to use for hydrological modelling by Chan et al. (2023a,b). Brunner and Slater (2022) apply a bias correction to simulated river flows, but highlight that, compared to observation-based estimates of extreme floods, their method can "introduce biases arising from the simulated meteorology and hydrological model".

### 5 Conclusions

The UNSEEN climate datasets provide a large ensemble of alternative historical climate realisations, allowing the direct sampling of more extreme meteorological events than the available observations, and a better assessment





of the likelihood of events (Thompson et al. 2017; Kelder et al. 2020). When combined with a simple hydrological
model, this similarly allows the direct sampling of more extreme hydrological events and better assessment of
likelihood. Both are conditional on a demonstration of fidelity for the event of interest.
An important issue for the hydrological modelling component is antecedent conditions. Here, two recent periods,
one very dry and one very wet, were selected as case studies to initialise and run the simple hydrological model
with the large ensemble of UNSEEN climate data. These case studies illustrate the potential of the approach to
assess the temporal evolution and spatial dependence of unprecedented but plausible hydrological extremes.
Clearly other periods could be similarly simulated and investigated, for example the summer 1976 drought, which
was one of the most extreme and extensive meteorological and hydrological droughts in recent history (Rudd et
al. 2017), and the widespread flooding of winter 2013/14, which was the wettest winter in Britian since records
began (Kay et al. 2018). The method could also be applied to other countries/regions, with an appropriate
hydrological model and using the same or similar global climate ensemble data.
Future work could include analysing the large-scale atmospheric drivers of selected hydrological extremes,
whether a record extreme for an individual month or persistently extreme for a number of months, which could
improve understanding of extreme events and their evolution. More detailed hydrological modelling, with (at
least) daily precipitation data, would ideally be used for flood case studies, when results could also be investigated
at a finer spatial scale, and additional hydrodynamic modelling (or pre-modelled design floods) could then provide
information on flood extents and impacts (e.g. Kay et al. 2018). Similarly, additional water resource system
modelling could provide information on drought impacts (e.g. Borgomeo et al. 2014). Soil moisture extremes
could also be investigated, with consequent impacts for agriculture as well as a range of natural hazards (e.g. Kay
et al. 2022), and other variables like groundwater levels could be investigated in a similar framework.
Being able to plan for unprecedented but plausible hydrological extremes is important in terms of improving the
resilience of water supply systems to drought (Chan et al. 2023a), and improving flood risk management and
incident response (Brunner and Slater 2022; Ganapathy et al. 2024). The UK water industry now has a statutory
obligation to demonstrate resilience to droughts that are more extreme than those recorded in the past, including
very rare events (e.g. 1:200 and 1:500 year) (Counsell and Durant, 2023). Similarly, for fluvial flood risk
management there is statutory requirement to plan for very extreme (rare) events that may lie outside observational
envelopes. In practice this is achieved through stochastic methods (e.g. using weather generators to produce long
precipitation series that are then run through hydrological and supply system models) or statistical methods
(pooling flood events from many catchments, or Probable Maximum Precipitation/Flood analyses). The UNSEEN
modelling chain described here provides a physically-informed alternative to complement these primarily
statistical approaches, with potential for use in both long-term water resource/flood risk planning and emergency
drought/flood response contexts.
The use of methods such as those presented here, deriving unprecedented events from historical case studies, can
aid preparedness by enabling planning for events similar to, but more extreme than, known events (with known
responses and impacts). Increasing resilience to potential extremes in the current climate will also provide some
resilience to the effects of climate change, which is expected to increase both floods and droughts in future in the
UK (e.g. Lane and Kay 2021; Kay et al. 2021; Rudd et al. 2019, 2023).
**Data availability.** The data are available from the authors upon reasonable request.





**Author contribution.** Conceptualisation and methodology: all authors. Formal analysis and visualisation: AK.
Funding acquisition: JH. Writing – original draft: AK. Writing – review and editing: all authors.
**Competing interests.** The authors declare that they have no conflict of interest.
**Acknowledgements.** This work was supported by the Natural Environment Research Council award number
NE/X019063/1 (Hydro-JULES). Thanks to UKCEH colleague Wilson Chan for comments on the draft
manuscript.

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
