# Peer review of "Demonstrating the use of UNSEEN climate data for hydrological"

_Natural Hazards and Earth System Sciences, 2024_

## Author Comment (AC1)

**Response to Review 1 of "Demonstrating the use of UNSEEN climate data for hydrological applications: case studies for extreme floods and droughts in England" by Kay et al.**

The authors would like to thank the reviewer for their comments on our manuscript. We describe below the changes we will make in response to each comment.

The study provides an application of the UNSEEN climate data sets to eight regions in England assessing the potential effects for one recent flood and one recent drought event using a modeling chain including a simple monthly water balance model informed by long historical run of the G2G model and including fidelity tests.
In my opinion, this is a valuable contribution to the discussion on how to be able to estimate of plausible, but yet unseen future extreme events.
The structure of the manuscript is logical and easy to follow and it is very clearly written overall.
Thank you

I have only minor comments and would else recommend publication:
- while for droughts and drought recovery in most cases a monthly temporal scale is sufficient, for floods daily or often sub-daily is the scale of interest. The authors mention in their conclusion that that is the case, but I would recommend to pick that up earlier in the manuscript, best already in the methods, maybe discussed in limitations again
We will add mention of this earlier in the manuscript, and refer the reader to Section 4.2 on 'limitations of the models and data', where it is addressed.

- in the description of the summer drought 2022 and autumn flood 2023 (2.3.1 and 2.3.2) how much was each region affected by these? all similar? Some particular?
We will add some more information about regional similarities/differences.

- the figure colors can not be distinguished if printed in b&w, please consider adjusting the hue.
We will revisit the colour scheme used in the figures, but given the complexity it may not be possible for all elements to show clearly in black and white.

Line by line comments (mostly editorial):
page 3
- L11: remove brackets
- L12: place the unit directly after rates
- L13: remove brackets
OK, thank you.

page 5
-L6 remove brackets; are these variables just examples or is the list exhaustive?
This will be clarified.

page 7
-L20 remove brackets
OK, thank you.

---

## Author Comment (AC2)

**Response to Review 2 of "Demonstrating the use of UNSEEN climate data for hydrological applications: case studies for extreme floods and droughts in England" by Kay et al.**

The authors would like to thank the reviewer for their comments on our manuscript. We describe below the changes we will make in response to each comment.

This paper provides a valuable demonstration of the application of coupled rainfall-flood model ensembles to extreme events through two UK studies. As the authors note in their Conclusions, this is a physically-informed alternative to primarily statistical existing methodologies, and in my opinion this is especially important. Overall it is an excellent, well written article and I would recommend publication once the Minor comments below are addressed.
Thank you

Some details of the methodology can be difficult to understand on a first read of the Methods section, but this is hard to avoid with so many datasets and different ensemble setups, and the authors have clearly endeavoured to try and lead the reader through. In a few places just a little extra elaboration on underpinning datasets or analogue methods (eg UKHO) could be considered.
We will try to provide further information where we think it may be helpful.

I note only minor corrections that are primarily editorial in character:

- Page 3, lines 16 – 24: the bias correction factors are not currently motivated initially. The downscaling felt obvious enough, but I was curious as to the specific need for this part. Only short motivation would be needed.

Typically the Hadley Centre global climate model (in common with climate models generally) show biases in the temporal pattern of precipitation through the year – the simple scheme applied here aims to correct these coarse biases in as simple a way as possible. We will add a sentence to the manuscript to motivate this part of the method.

- Page 3, line 28/Fig1b: would be useful to provide some physical interpretation of the SAAR ratios, thus rationalising their use in the downscaling. For example, from fig 1b they appear to in part track orography (which one would expect), but also to significantly deviate. Please expand a bit on what these ratios physically mean.

Yes, orography is a major part of this, but also things like rain-shadow effect (using nearby topographic shape) are included in the original SAAR derivation (Perry and Hollis 2005, Int. J. Climatol. 25: 1023–1039). Note that we are interested in the relative values of 1km SAAR to grid-box mean SAAR within each grid-box, hence the clear changes at the grid-box edges. A sentence will be added providing further explanation.

- Page 5, line 16: are these the same 17 regions as in Fig.1, and of which the regions used in the study are a subset?

Yes, these are the same regions shown in Fig1c – this will be clarified.

- Page 5, line 17: significance of correlation?

Unfortunately the cited work does not provide information on correlation significance.

- Page 5, lines 29 – 31: please expand on how WBM Obs generates an ensemble. I presume the members are the different years of HadUK-Grid to 41 members in all, each initialised from case studies' antecedent conditions? As a reader not familiar with the HadUK-Grid dataset or Ensemble Streamflow method, the current exposition isn't quite

enough to piece together how the Obs ensemble is built (which is important for interpreting results).

Yes, all years of HadUK-Grid data (1961-2022) are used for each case study (regardless of the year of the case study) – this will be clarified.

- Page 9, table 2: given the comments made on the February fidelity failures for single statistics, its striking to not have a comment on the August SEEngland failure of multiple statistics! Why was this?
  - o Failure of multiple tests is acknowledged in Page 12 line 28-29; but not explained.

We will add something about possible reasons for the August SEEngland result.

- Figs 2/3: no units on graphs, please correct. It's a shame that arguably the most important information on these plots is compressed to the RHS – is several years' preceding data necessary? I appreciate adds context to the extreme flows.

The regional mean flows are standardised, hence no units. Yes, we appreciate that the ensemble results are somewhat compressed on these two figures, but felt that showing the context of the variability in flows in the recent past was important. Later figures focus on the event periods.

- Figs 4/5: worth explicitly noting that the order of members in Fig legend corresponds to month when said member is most extreme. Not immediate which month each member is relevant to, which is maybe more important that the initialisation batch?

This information will be added.

- Page 12, figs 4/5: not necessarily a correction, but note I find it difficult to see the spatial connections between member extremes. Authors have spelled this out in text, but current visualisation does not make awfully clear.

The spatial connection is just done by comparing the legends between the regions, to see where ensemble members are duplicated between regions. Most members are not duplicated at all between regions, and we could not think of a good way to highlight those that are other than describing them in the main text.

- Figs 6a/7a: the difference between UNSEEN and Obs is difficult to see (appreciate slightly the point in Recovery). However a log scale may be useful to draw out differences which are there, highlighting regions where forecasts and Obs differ notably?

We will try out a log-scale as suggested, to see if this does help to highlight any differences.

- Page 16, line 5: please improve the clarity of wording here, i.e. to emphasise you are now referring to WBM Obs extremes which are not present in UNSEEN ensemble (at least, I believe that is the point).

This will be clarified.

- Page 18, line 4: worth redefining AE/PE acronyms. Not used since Methods, and as a meteorologist rather than hydrologist their meaning is not immediately obvious.

This will be clarified.

---

## Author Response (AR1)

**Response to Reviews of "Demonstrating the use of UNSEEN climate data for hydrological applications: case studies for extreme floods and droughts in England" by Kay et al.**

**Reviewer 1**

The study provides an application of the UNSEEN climate data sets to eight regions in England assessing the potential effects for one recent flood and one recent drought event using a modeling chain including a simple monthly water balance model informed by long historical run of the G2G model and including fidelity tests.
In my opinion, this is a valuable contribution to the discussion on how to be able to estimate of plausible, but yet unseen future extreme events.
The structure of the manuscript is logical and easy to follow and it is very clearly written overall.
Thank you

I have only minor comments and would else recommend publication:
- while for droughts and drought recovery in most cases a monthly temporal scale is sufficient, for floods daily or often sub-daily is the scale of interest. The authors mention in their conclusion that that is the case, but I would recommend to pick that up earlier in the manuscript, best already in the methods, maybe discussed in limitations again
We have added mention of this in Section 2.3, and referred the reader to Section 4.2 on 'limitations of the models and data' where it was already discussed.

- in the description of the summer drought 2022 and autumn flood 2023 (2.3.1 and 2.3.2) how much was each region affected by these? all similar? Some particular?
We have added some more information about regional differences in conditions (Sections 2.3.1 and 2.3.2).

- the figure colors can not be distinguished if printed in b&w, please consider adjusting the hue.
The colours used for the historical ranges in Figs 2-5 are as used in the UKHO, so have been retained here. We have adjusted the width of the set of lines indicating the WBM UNSEEN range in Figs 2 and 3, to try to distinguish it better from the WBM Obs range even if in black and white. The line colours in Figs 4 and 5 are more incidental, as they can be distinguished by the month of the extreme occurrence and line type/width.

Line by line comments (mostly editorial):
page 3
- L11: remove brackets
- L12: place the unit directly after rates
- L13: remove brackets
OK, thank you.

page 5
-L6 remove brackets; are these variables just examples or is the list exhaustive?
This has been clarified (Section 2.2).

page 7
-L20 remove brackets
We prefer to keep the brackets here.

**Reviewer 2**

This paper provides a valuable demonstration of the application of coupled rainfall-flood model ensembles to extreme events through two UK studies. As the authors note in their Conclusions, this is a physically-informed alternative to primarily statistical existing methodologies, and in my opinion this is especially important. Overall it is an excellent, well written article and I would recommend publication once the Minor comments below are addressed.
Thank you

Some details of the methodology can be difficult to understand on a first read of the Methods section, but this is hard to avoid with so many datasets and different ensemble setups, and the authors have clearly endeavoured to try and lead the reader through. In a few places just a little extra elaboration on underpinning datasets or analogue methods (eg UKHO) could be considered.
We have added further information in a number of places, particularly expanding on UKHO background.

I note only minor corrections that are primarily editorial in character:
- Page 3, lines 16 – 24: the bias correction factors are not currently motivated initially. The downscaling felt obvious enough, but I was curious as to the specific need for this part. Only short motivation would be needed.
We have added some motivation for the bias-correction of GCM rainfall data (Section 2.1).

- Page 3, line 28/Fig1b: would be useful to provide some physical interpretation of the SAAR ratios, thus rationalising their use in the downscaling. For example, from fig 1b they appear to in part track orography (which one would expect), but also to significantly deviate. Please expand a bit on what these ratios physically mean.
We have added some discussion on the meaning of the SAAR ratios (Section 2.1).

- Page 5, line 16: are these the same 17 regions as in Fig.1, and of which the regions used in the study are a subset?
Yes; this has been clarified (Section 2.2).

- Page 5, line 17: significance of correlation?
Unfortunately the cited work does not provide information on correlation significance.

- Page 5, lines 29 – 31: please expand on how WBM Obs generates an ensemble. I presume the members are the different years of HadUK-Grid to 41 members in all, each initialised from case studies' antecedent conditions? As a reader not familiar with the HadUK-Grid dataset or Ensemble Streamflow method, the current exposition isn't quite enough to piece together how the Obs ensemble is built (which is important for interpreting results).
Yes, all years of HadUK-Grid data (1961-2022) are used for each case study (regardless of the year of the case study) – this has been clarified (Section 2.3).

- Page 9, table 2: given the comments made on the February fidelity failures for single statistics, its striking to not have a comment on the August SEEngland failure of multiple statistics! Why was this?
  o Failure of multiple tests is acknowledged in Page 12 line 28-29; but not explained.
We have added a sentence about the August SE England failure, which is for both skewness and kurtosis (Section 3.1).

- Figs 2/3: no units on graphs, please correct. It's a shame that arguably the most important information on these plots is compressed to the RHS – is several years' preceding data necessary? I appreciate adds context to the extreme flows.

The regional mean flows are standardised, hence no units. Yes, we appreciate that the ensemble results are somewhat compressed on these two figures, but felt that showing the context of the variability in flows in the recent past was important. Later figures focus on the event periods.

- Figs 4/5: worth explicitly noting that the order of members in Fig legend corresponds to month when said member is most extreme. Not immediate which month each member is relevant to, which is maybe more important that the initialisation batch?

This has been clarified (captions of Fig 4 and 5).

- Page 12, figs 4/5: not necessarily a correction, but note I find it difficult to see the spatial connections between member extremes. Authors have spelled this out in text, but current visualisation does not make awfully clear.

The spatial connection is just done by comparing the legends between the regions, to see where ensemble members are duplicated between regions. Most members are not duplicated at all between regions, and we could not think of a good way to highlight those that are other than describing them in the main text.

- Figs 6a/7a: the difference between UNSEEN and Obs is difficult to see (appreciate slightly the point in Recovery). However a log scale may be useful to draw out differences which are there, highlighting regions where forecasts and Obs differ notably?

We tried a log-scale but it doesn't help to highlight the differences; in fact it's worse unless ymin is set quite high, which we feel could be misleading.

- Page 16, line 5: please improve the clarity of wording here, i.e. to emphasise you are now referring to WBM Obs extremes which are not present in UNSEEN ensemble (at least, I believe that is the point).

This has been clarified (Section 3.4).

- Page 18, line 4: worth redefining AE/PE acronyms. Not used since Methods, and as a meteorologist rather than hydrologist their meaning is not immediately obvious.

Done (Section 4.2).